# Grp78 is required for intestinal Kras-dependent glycolysis proliferation and adenomagenesis

Claudia N Spaan[1,2,5],*, Ruben J de Boer[1,2,5],*, Wouter L Smit[1,2,5], Jonathan HM van der Meer[1,5], Manon van Roest[1,5], Jacqueline LM Vermeulen[1,5], Pim J Koelink[1,5], Marte AJ Becker[1,5], Simei Go[1,5], Joana Silva[3], William J Faller[3], Gijs R van den Brink[1], Vanesa Muncan[1,5], Jarom Heijmans[1,2,4,5]

In development of colorectal cancer, mutations in *APC* are often followed by mutations in oncogene *KRAS*. The latter changes cellular metabolism and is associated with the Warburg phenomenon. Glucose-regulated protein 78 (*Grp78*) is an important regulator of the protein-folding machinery, involved in processing and localization of transmembrane proteins. We hypothesize that targeting *Grp78* in *Apc* and *Kras* (*AK*)-mutant intestines interferes with the metabolic phenotype imposed by *Kras* mutations. In mice with intestinal epithelial mutations in *Apc*, *Kras^{G12D}* and heterozygosity for *Grp78* (*AK-Grp78^{HET}*) adenoma number and size is decreased compared with *AK-Grp78^{WT}* mice. Organoids from *AK-Grp78^{WT}* mice exhibited a glycolysis metabolism which was completely rescued by *Grp78* heterozygosity. Expression and correct localization of glucose transporter GLUT1 was diminished in *AK-Grp78^{HET}* cells. GLUT1 inhibition restrained the increased growth observed in *AK*-mutant organoids, whereas *AK-Grp78^{HET}* organoids were unaffected. We identify *Grp78* as a critical factor in *Kras*-mutated adenomagenesis. This can be attributed to a critical role for *Grp78* in GLUT1 expression and localization, targeting glycolysis and the Warburg effect.

## Introduction

Colorectal cancer (CRC) is one of the leading causes of cancer-related death in Western countries (Global Burden of Disease Cancer Collaboration, 2017). Development of CRC results from accumulation of a distinct set of mutations causing stepwise development from healthy tissue to adenomas and, subsequently, carcinomas, a sequence known as the adenoma-to-carcinoma sequence (Vogelstein et al, 1988; Fearon & Vogelstein, 1990).

Activating mutations in the *KRAS* gene are among the major oncogenic drivers and occur as a second hit after *APC* mutations. *KRAS* is mutated in over 40% of all CRC cases and in CRCs that have no mutations in *KRAS*, mutations in *BRAF* may occur (Bos et al, 1987; Forrester et al, 1987; Samowitz et al, 2005; Normanno et al, 2009; Ogino et al, 2009).

*KRAS* mutations succeed loss of tumor suppressor gene *APC* or alternatively, after activating mutations in *β-catenin* (Forrester et al, 1987; Rubinfeld et al, 1993; Su et al, 1993). Sole mutations in *KRAS*, which do not arise in the context of prior *WNT*-activating mutations are exceedingly rare in adenomas, and in a murine model, a single *Kras* mutation resulted in senescence as opposed to hyperproliferation seen in the *Apc-Kras* mutational sequence (Bennecke et al, 2010; Feng et al, 2011; Smit et al, 2020). In colorectal adenomas, occurrence of a *KRAS* mutation on top of *WNT* signal-activating mutations was associated with the transition of small to large adenomas (Fearon & Vogelstein, 1990; Feng et al, 2011).

In the murine process of adenoma development, *Apc* and *Kras* synergistically stimulate proliferation and stemness and usher development of small adenomas into larger adenomas (Fearon & Vogelstein, 1990; Janssen et al, 2006; Sakai et al, 2018). To meet the increased demand for cellular growth, *Kras*-mutant cells have a higher protein translation capacity and an altered metabolic profile (Kimmelman, 2015). The molecular CRC subtype CMS3 that relies on activated (and frequently mutated) *KRAS* is also characterized by high metabolic activity (Guinney et al, 2015). In addition, we and others have shown that *Kras* potentiates global protein translation in organoids with homozygous loss of *Apc* (Smit et al, 2020; Knight et al, 2021). Furthermore, mutant *KRAS* promotes the expression of glucose uptake receptor GLUT1 and thereby increases glycolysis (Racker et al, 1985; Yun et al, 2009; Ying et al, 2012). In line with this, restoring *KRAS* to WT in CRC cells reduces glycolysis and growth. *Kras*-driven tumors are thus dependent on glycolysis and targeting

[1]Department of Gastroenterology and Hepatology, Tytgat Institute for Liver and Intestinal Research, Amsterdam UMC, University of Amsterdam, Amsterdam, Netherlands [2]Cancer Center Amsterdam, Cancer Biology and Immunology, Amsterdam, Netherlands [3]Department of Oncogenomics, Netherlands Cancer Institute, Amsterdam, Netherlands [4]Department of Internal Medicine, Amsterdam UMC, University of Amsterdam, Amsterdam, Netherlands [5]Amsterdam Gastroenterology Endocrinology Metabolism, Amsterdam University Medical Centers, Amsterdam, Netherlands

Correspondence: j.heijmans@amsterdammumc.nl; r.j.deboer@amsterdammumc.nl
Gijs R van den Brink's present address is Roche Innovation Center Basel, F Hoffmann-La Roche AG, Basel, Switzerland
*Claudia N Spaan and Ruben J de Boer contributed equally to this work

the glycolytic pathway decreases tumor proliferation in preclinical studies (Xie et al, 2014; Sheng & Tang, 2016).

In previous work, we have pinpointed a role for the endoplasmic reticulum (ER)-resident chaperone glucose-regulated protein 78 (*Grp78*) as a potential target to reduce intestinal adenoma formation in mice. *Grp78* is a critical mediator of the unfolded protein response (UPR) and activation of the UPR, as seen after deletion of *Grp78*, leads to temporary inhibition of protein translation and increases the ER size (Ma & Hendershot, 2001; Harding et al, 2002; Hetz, 2012). In addition, UPR activation results in increased transcription and translation of *Grp78* in a stimulatory feedback loop. Deletion of a single copy of *Grp78* results in haploinsufficiency, in which the threshold for activation of the UPR is lowered (van Lidth de Jeude et al, 2018; Rangel et al, 2021). In *Apc*-heterozygous mice, that over time develop intestinal adenomas, *Grp78* heterozygosity was associated with reduced adenoma-genesis, and organoids of these mice exhibited reduced protein translation (van Lidth de Jeude et al, 2018).

Besides its role in the UPR and protein synthesis, GRP78 is known as a glucose-sensing protein (Li et al, 2015). Up-regulation of GRP78 in solid tumors results from poor vascularization and glucose deprivation and GRP78 up-regulation results in increased expression of glucose transporter GLUT1 (Toyoda et al, 2018). GLUT1 and GRP78 are reported to be up-regulated in several cancers and a high GLUT1 expression corresponds with a poor prognosis in CRC (Yamamoto et al, 1990; Haber et al, 1998; Li & Lee, 2006).

We hypothesized that *Grp78* heterozygosity would challenge CRC growth, because of altered protein translation and glucose consumption. This would result in inhibition of growth, specifically in the context of *Apc* and *Kras* mutations compared with a single *Apc* mutation. We thus set out to determine the effect of *Grp78* haploinsufficiency in *Apc-Kras* mutant intestinal adenomagenesis.

## Results

### Grp78 heterozygosity reduces colon adenoma initiation and progression in Apc-Kras^G12D mice

We modeled human disease by generating mice that harbored an inducible deletion in *Apc* and mutant *Kras^G12D*. *Apc* and *Kras* mutations synergize to augment the number and size of intestinal adenomas and adenoma localization is shifted towards the colon (Sakai et al, 2018). All mice were crossed into *VillinCre^ERT2* mice and Cre-mediated recombination occurred in the small intestinal and colonic epithelia after tamoxifen injections.

*VillinCre^ERT2-Apc^+/fl-Kras^G12D/+* mice, were crossed into *Grp78* heterozygously floxed mice (further referred to as *AK-Grp78^HET*) and *Grp78* WT littermates were used as control animals (*AK-Grp78^WT*). To assess the contribution of the mutant *Kras* allele, *VillinCre^ERT2-Apc^+/fl-Grp78^+/+* mice (further referred to as *A-Grp78^WT*) were used as comparison.

To confirm recombination of *Grp78* throughout the intestine, we combined GRP78 immunohistochemistry with specific visualization of the novel *Grp78^Δ5−7* by *Basescope* (Fig 1A). As expected,

immunostaining for GRP78 was present in *AK-Grp78^HET* mice, resulting from staining of the protein product of the remaining *Grp78* allele. In contrast, because of up-regulation of *Grp78^Δ5−7 mRNA* transcription in *AK-Grp78^HET* mice, specific expression of the floxed *Grp78^Δ5−7 mRNA* was abundant. In addition, we confirmed recombination efficacy of the *Kras* allele throughout the intestine by immunohistochemistry for the mutant KRAS (Fig 1A). In addition, β-catenin was highly expressed as a consequence of mutated *Apc* (Fig 1A). Next, we analyzed GRP78 protein expression in epithelial cells from *AK-Grp78^WT* and *AK-Grp78^HET* mice. To this end, we generated organoids from mice with these genotypes. To investigate the specific role of *Grp78* in combination with *Kras*, we compared these organoid with lines without mutant *Kras* (*A-Grp78^WT* and *A-Grp78^HET*) (van Lidth de Jeude et al, 2018). The resulting *Grp78^Δ5−7* gene product has an open reading frame that is out of frame, resulting in no detectable protein being made. Thus, GRP78 protein from *Grp78^HET* cells reflects expression from the single WT allele in these cells (Luo et al, 2006). Using immunoblots, we confirmed that GRP78 protein levels were diminished in *AK-Grp78^HET* compared with *AK-Grp78^WT* organoids (Fig 1B).

In mice of both *AK-Grp78^WT* and *AK-Grp78^HET* genotypes, we investigated cell fate by analysis of proliferation and apoptosis. Using a 2-h BrdU incorporation pulse, we found that enterocytes of *AK-Grp78^HET* animals exhibited modestly reduced proliferation compared with their WT counterparts (8.3 versus 6.5 $P < 0.05$, Fig 1C and D). Apoptosis, visualized by cleaved caspase immunohistochemistry, was not significantly reduced (13.8 versus 9.2 cells per 100 crypts, $P = 0.09$) (Fig S1A). From the moment of recombination, both *AK-Grp78^WT* and *AK-Grp78^HET* animals lost weight compared with *A-Grp78^WT* animals, but weight did not differ between *AK-Grp78^HET* and *AK-Grp78^WT* animals (Fig S1B). Of all *AK-Grp78^WT* mice, 80% developed a rectal prolapse, indicating severe rectal adenoma formation, whereas *AK-Grp78^HET* mice were markedly protected from development of rectal prolapse (20%, $P = 0.054$, Fig 1E and F) (Colnot et al, 2004). As expected, colons of *A-Grp78^WT* mice contained no adenomas (Fig 1G). In the colon, *AK-Grp78^WT* and *AK-Grp78^HET* animals harbored multiple adenomas (Fig 1G–I). *AK-Grp78^WT* animals exhibited markedly increased adenoma numbers and growth compared with *AK-Grp78^HET* animals (total adenomas [*of which large*] = 61.4 [*31.6*] versus 40.6 [*17*], $P < 0.01$, Fig 1I and J). In the small intestine, adenoma numbers were very low and there was a nonsignificantly reduced adenoma number in *AK-Grp78^HET* animals (Fig S1C and D).

The above results display that *Grp78* levels are important for initiation and growth of *Apc* and *Kras* mutant adenomas, especially in the large intestine.

### Grp78 levels are critical for proliferation, protein production, and stemness

To analyze how adenomagenesis was reduced in *AK-Grp78^HET* mice, we further analyzed intestinal epithelial organoids of *AK-Grp78^WT* and *AK-Grp78^HET* genotypes. After recombination, increased growth of *AK-Grp78^WT* organoids was observed as expected (Fig 2A). However, *AK-Grp78^HET* organoids did not increase in size, suggesting that *Grp78* is a rate-limiting factor, responsible for growth upon

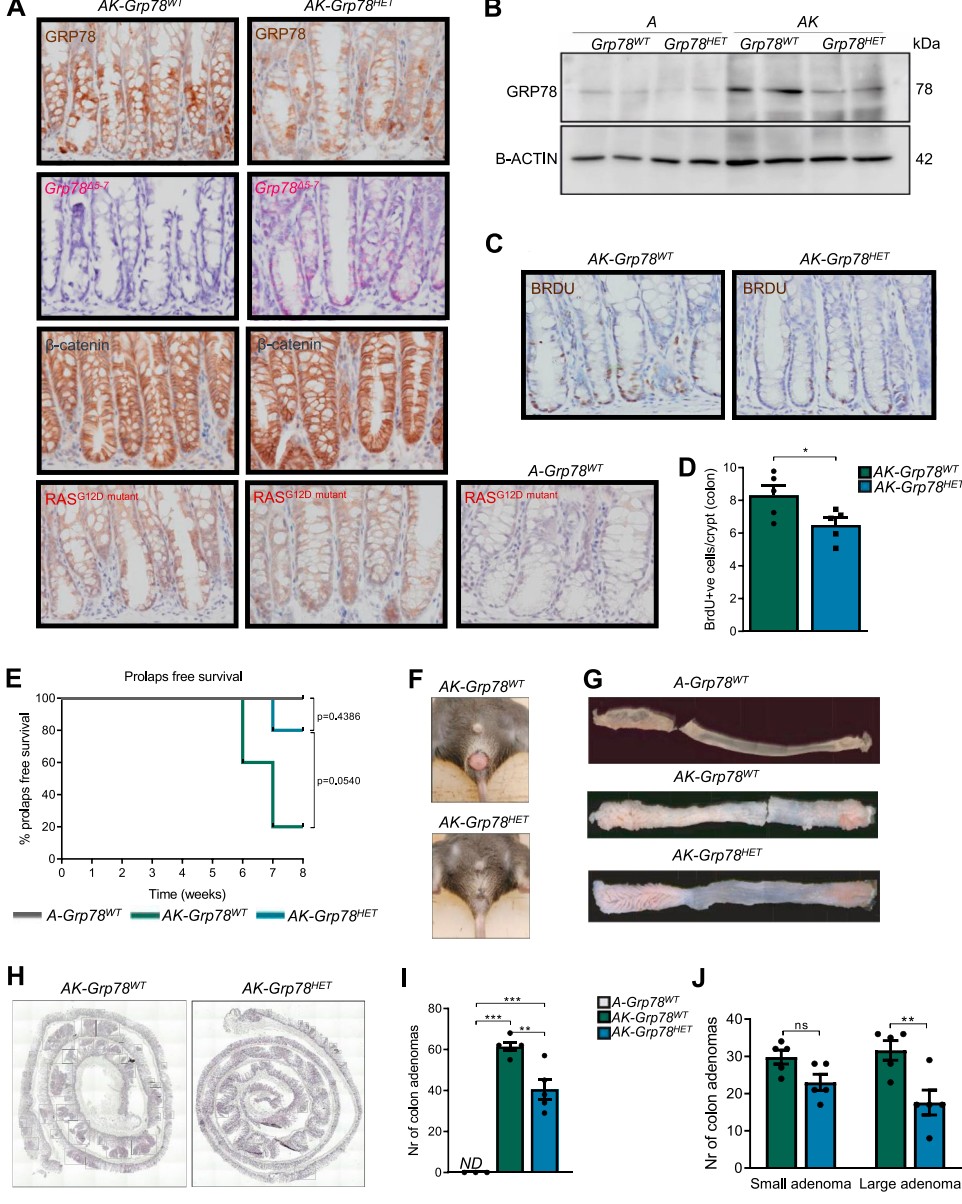

**Figure 1.  *Grp78* heterozygosity reduces colon adenoma initiation and progression in *Apc-Kras^G12D* mice.**
**(A)** Representative images showing the expression of GRP78, KRAS^G12D, β-catenin by immunohistochemistry and representative images from *mRNA* in situ hybridization of the floxed *Grp78^Δ5−7* transcript n consecutive sections. **(B)** Immunoblot for GRP78 on organoids from indicated genotypes. **(C)** Representative BrdU staining of indicated genotypes (*AK-Grp78^WT*, *AK-Grp78^HET*). **(D)** Quantification of BrdU staining in crypts of colon in indicated genotypes. (*AK-Grp78^WT* N = 5, *AK-Grp78^HET* N = 5). **(E)** Kaplan–Meier curve of prolapse-free survival (*A-Grp78^WT* N = 3, *AK-Grp78^WT* N = 5, *AK-Grp78^HET* N = 5). **(F)** Representative images of rectal prolapse. **(G)** Representative whole-mount images of colons. **(H)** Representative H&E images of entire colons, adenomas are indicated by boxes. **(I)** Number of colon adenomas (*A-Grp78^WT* N = 3, *AK-Grp78^WT* N = 5, *AK-Grp78^HET* N = 5). **(J)** Number of colon adenomas separated by size. Small adenoma = <3 crypts, large adenoma = ≥3 crypts (*A-Grp78^WT* N = 3, *AK-Grp78^WT* N = 5, *AK-Grp78^HET* N = 5). *$P < 0.05$, **$P < 0.01$, ***$P < 0.001$; ns, non-significant; *ND*, Not detectable.

accumulating oncogenic mutations in *Apc* and *Kras* (Fig 2A and B). We next analyzed proliferation by EdU incorporation and quantified organoid size. Interestingly, the increased growth that was observed in *AK-Grp78^WT* versus *A-Grp78^WT* was almost completely nullified in *AK-Grp78^HET* organoids (Fig 2C–E).

Heterozygosity of master ER chaperone *Grp78* decreases the threshold for ER stress, causing up-regulation of ER chaperones *Grp78* and *Grp94*, thereby regulating constant levels of *Grp78 mRNA* (Bertolotti et al, 2000). Using *mRNA* primers that did not distinguish between WT and floxed *mRNA* transcripts, we indeed observed unaltered total *Grp78 mRNA* expression levels in *AK-Grp78^HET* organoids compared with *AK-Grp78^WT* organoids, whereas *Grp94 mRNA* was increased (Fig S2A and B). To confirm deletion of a single *Grp78 allele*, we used a specific primerset to measure expression of the truncated (incapacitated) *Grp78* transcript derived from the

*Grp78^Δ5−7*-floxed allele (van Lidth de Jeude et al, 2018). The floxed *Grp78^Δ5−7* was only observed in in the *AK-Grp78^HET* organoids, confirming the loss of a single allele of *Grp78* (Fig 2F).

We recently reported a significant rise in global protein translation when *Apc*-mutant cells acquire a *Kras* mutation (Smit et al, 2020). We therefore tested whether reduced GRP78 levels in *AK-Grp78^HET* organoids would negatively impact protein translation. Indeed, translation capacity was significantly increased in *AK-Grp78^WT* organoids compared with *A-Grp78^WT* organoids (Fig 2G). Strikingly, the increased protein translation that was observed in *AK-Grp78^WT* crypts was largely (52%) reduced in *AK-Grp78^HET* organoids ($P < 0.01$).

Thus, *AK-Grp78^HET* organoids exhibited reduced size, proliferation, and protein translation. We next analyzed whether these alterations coincided with reduced stemness. Previously, we

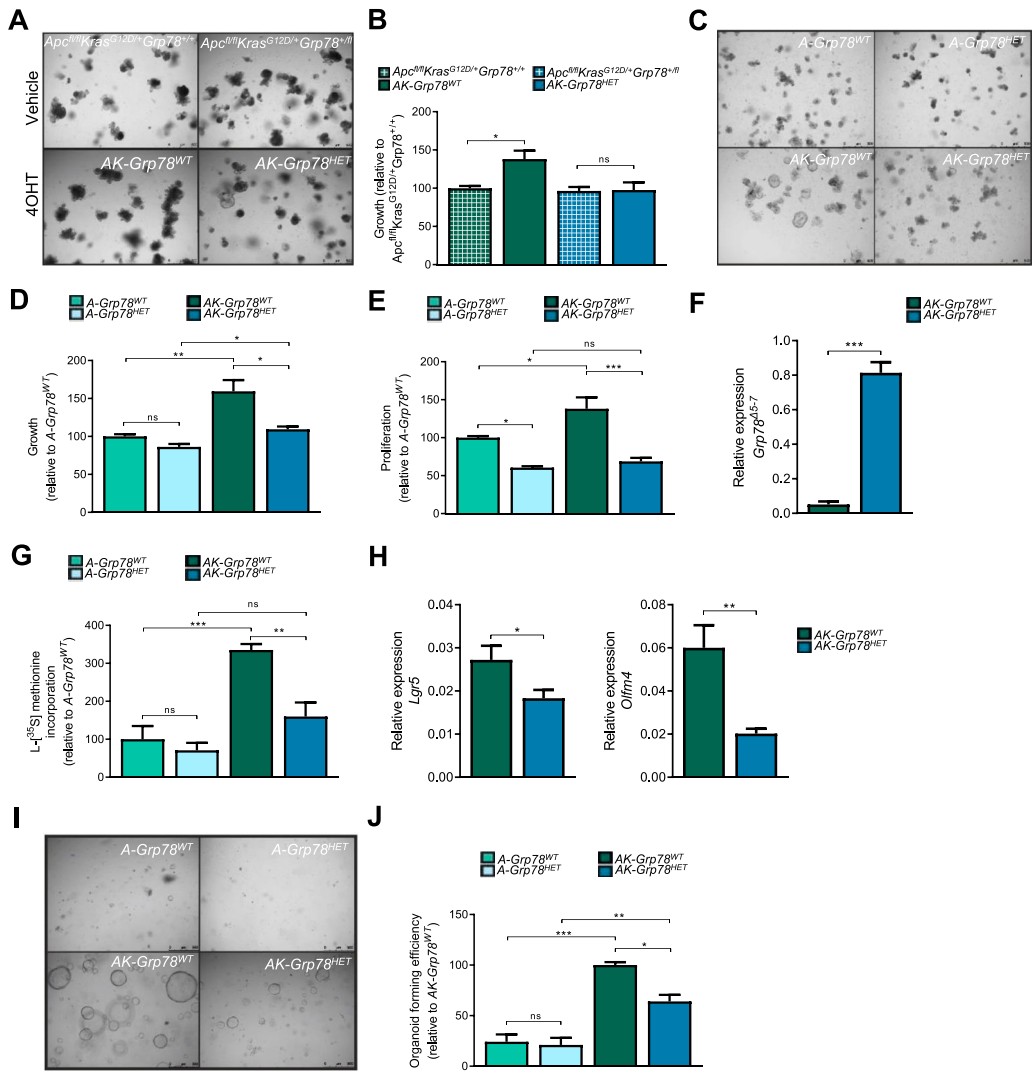

**Figure 2.  *Grp78* is critical for *Kras*-dependent growth in the context of *Apc* mutant organoids.**
**(A)** Representative bright-field images of *Apc*^fl/fl^*Kras*^G12D/+^*Grp78*^+/+^ and *Apc*^fl/fl^*Kras*^G12D/+^*Grp78*^+/fl^ small intestinal organoids upon recombination with 4-OHT (AK-*Grp78*^WT^ and AK-*Grp78*^HET^) or treatment with the vehicle. **(B)** Organoid growth measured by the surface area of *Apc*^fl/fl^*Kras*^G12D/+^*Grp78*^+/+^ and *Apc*^fl/fl^*Kras*^G12D/+^*Grp78*^+/fl^ organoids upon recombination with 4OHT (*AK-Grp78*^WT^ and *AK-Grp78*^HET^) or treatment with the vehicle. N = 3 per genotype. **(C)** Representative bright-field images of *A-Grp78*^WT^, *A-Grp78*^HET^, *AK-Grp78*^WT^, and *AK-Grp78*^HET^ organoids. Mice with genotype *VillinCre*^ERT2,^ *Apc*^fl/fl^, *Kras*^G12D/+^ or ^+/+^ and *Grp78*^+/+^ or ^+/fl^ were crossed to gain four different types of organoids. **(D)** Growth of organoids 72 h after induction of the genotypes. Quantification of area relative to the control *A-Grp78*^WT^. **(E)** Proliferation measured by EdU incorporation relative to the control *A-Grp78*^WT^. **(F)** Quantitative RT–PCR analysis with specific primers for the *Grp78*^Δ5–7^ *mRNA*. **(G)** Global protein translation measured with ^[35S]^methionine incorporation in organoids from the indicated genotypes. **(H)** Quantitative RT–PCR analysis for *Lgr5* and *Olfm*4 in organoids from the indicated genotypes. **(I)** Bright-field image of organoids grown after single-cell seeding. **(J)** Quantification of organoids grown from equal numbers of single cells from indicated genotypes. Quantification of area relative to the control *AK-Grp78*^WT^. *P < 0.05, **P < 0.01, ***P < 0.001; ns, nonsignificant.

found reduced *mRNA* expression of stem cell markers in *A-Grp78*^HET^ organoids compared with *A-Grp78*^WT^ organoids (van Lidth de Jeude et al, 2018). Moreover, adenoma progression from the *Apc* mutant to *Apc-Kras* double-mutant state, is marked by an increase in stemness (Janssen et al, 2006). In *AK-Grp78*^HET^ organoids, a significant decrease in transcription of intestinal stem cell markers *Lgr5* and *Olfm4* was observed compared with *AK-Grp78*^WT^ organoids (Fig 2H). Thus, *Grp78* remains critical for the increase in stemness that is observed in *Apc*-mutant neoplastic crypts and *Grp78* is a rate-limiting factor during progression from *Apc* to *Apc-Kras* double-mutant organoids. To examine stem cell clonogenicity on a functional level, we performed single-cell seeding experiments. Single cells without an oncogenic *Kras*^G12D^ mutation exhibited a limited self-renewal capacity and hardly grew into organoids (Fig 2I and J). In agreement with reduced *mRNA* expression of stem cell markers, *AK-Grp78*^HET^ cells had a significantly reduced capacity for organoid outgrowth (64 versus 100, *P* < 0.05) and grew into smaller organoids than *AK-Grp78*^WT^ controls (Fig 2I).

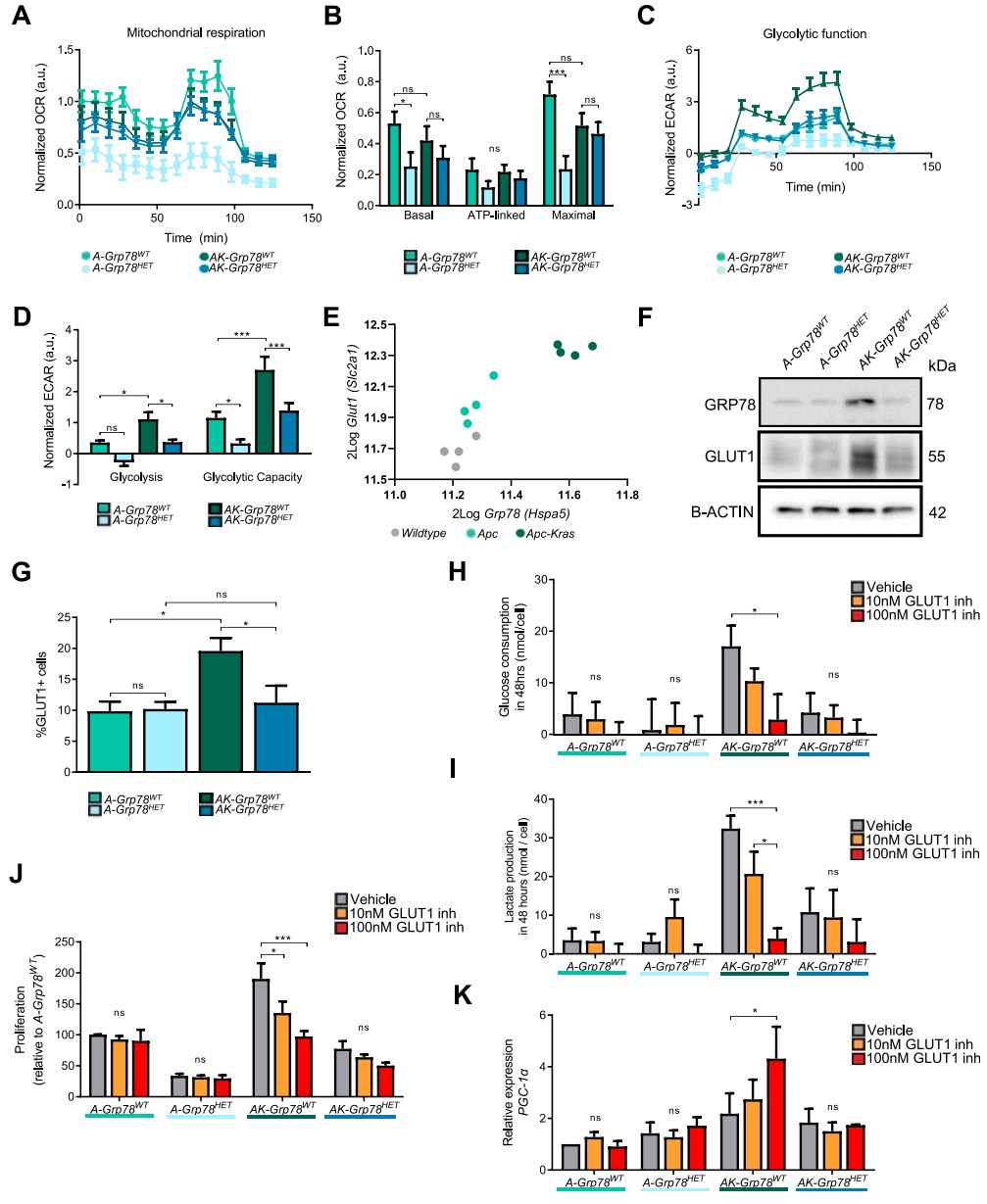

**Figure 3. GLUT1-dependent glycolysis in *Apc-Kras* mutant organoids is regulated by *Grp78*.**
**(A)** Oxygen consumption rate in organoids of indicated genotypes, measured by Agilent Seahorse. **(B)** Basal respiration, ATP-linked respiration, and maximum respiration in the indicated organoids. **(C)** Glycolytic function in the organoids, measured by Agilent Seahorse. **(D)** Glycolysis and glycolytic capacity in the indicated organoids. **(E)** *Grp78 mRNA* and *Glut1 mRNA* expressions derived from microarray analysis of indicated mouse organoids (GSE143509). **(F)** Immunoblots of GRP78 and GLUT1 from indicated organoids. **(G)** Flow cytometric immunostaining of membrane-localised GLUT1 protein on live cells, percentage Alexa 647+ve of the parent (single cells, PI negative). **(H)** Glucose consumption upon treatment with vehicle, 10 , and 100 nM of GLUT1-inhibitor BAY-876. **(I)** Lactate production upon treatment with vehicle, 10 , and 100 nM of GLUT1-inhibitor BAY-876. **(J)** EdU incorporation upon treatment with vehicle, 10 , and 100 nM of GLUT1-inhibitor BAY-876. **(K)** Quantitative RT–PCR analysis for *Pgc1a mRNA* on organoids of indicated genotypes upon treatment with vehicle, 10 , and 100 nM of GLUT1-inhibitor BAY-876. $*P < 0.05$, $**P < 0.01$, $***P < 0.001$, ns, nonsignificant.

In the context of an additional *Kras^{G12D}* mutation, we find that *Grp78* is required for growth, stemness, and protein translation. *Grp78* heterozygosity almost completely abrogates the effect of a *Kras* mutation on growth and translation.

### Grp78-dependent glycolysis and proliferation in AK organoids is fueled by Glut1-mediated glucose influx

Under normal culture conditions, we noticed that growth medium acidified more rapidly in *AK-Grp78^{WT}* organoids compared with *AK-Grp78^{HET}* organoids. Acidification may be the result of increased glycolysis and consequent production of lactic acid under aerobic conditions. CRC cell lines with mutant *KRAS* depend on glycolysis (ECAR), a phenomenon known as the Warburg effect (Warburg,

1956). It has been shown that restoration of WT *KRAS* reverts the glycolytic Warburg effect towards metabolism in which oxidative phosphorylation (OCR) is dominant (Yun et al, 2009). Thus mutations in *Kras* that lead to constitutive *Kras* activation may impose a metabolic switch towards the Warburg effect. We next analyzed the influence of *Grp78* levels on *Kras*-induced Warburg metabolism. *A-Grp78^{WT}* mutant organoids had high levels of oxidative phosphorylation compared with organoids of all other genotypes (Fig 3A and B). In *A-Grp78^{HET}* organoids, both oxidative phosphorylation and glycolysis were reduced, which correlates with reduced growth and energy expenditure (Fig 3A–D). Moreover, stem cells are known to rely on oxidative phosphorylation and the observed reduction could deflect reduced stem cell numbers (Rodriguez-Colman et al, 2017). Upon introduction of the *Kras* mutation in *AK-Grp78^{WT}*

organoids, we could corroborate previous data that revealed that mutations in *Kras* introduce the metabolic switch which results in a transition from oxidative phosphorylation into predominant gly-colysis (Yun et al, 2009; Ying et al, 2012). Interestingly, *AK-Grp78$^{HET}$* organoids had comparable levels of oxidative phosphorylation with *AK-Grp78$^{WT}$* organoids, although glycolysis and glycolytic capacity were markedly reduced (Fig 3A–D). Glycolytic function in *AK-Grp78$^{HET}$* organoids was comparable with *A-Grp78$^{WT}$* organoids, suggesting sufficient *Grp78* is critical for *Apc*-deficient cells to undergo the glycolytic switch towards the Warburg effect after mutation of *Kras* (Fig 3C and D). We thus pinpoint *Grp78* as a critical factor for *Kras*-induced Warburg metabolism.

To further understand the molecular basis of *Grp78*-dependent glycolysis in *AK-Grp78$^{HET}$* organoids, we analyzed glucose influx. It has been shown that *Kras*-mutant CRC cells exhibit increased levels of one of the most important glucose transporters GLUT1 and depend on GLUT1 for their proliferative capacity (Yun et al, 2009). As knockdown of *Grp78* has been shown to reduce *Glut1 mRNA* levels, we hypothesized that reduced glycolysis in *AK-Grp78$^{HET}$* organoids may result from an inability of these organoids to induce sufficient levels of *Glut1* (Li et al, 2015). We analyzed both *Grp78* and *Glut1 mRNA* expression from our previously published *mRNA* microarray experiments (GSE143509) in *Apc*-mutant versus *Apc-Kras*-mutant organoids (Smit et al, 2020). Interestingly, we observed a gradual increase in *Glut1 mRNA* expression upon acquisition of *Apc* and *Kras$^{G12D}$* mutations, respectively, which correlated with *Grp78* expression (r = 0.92; P < 0.001) (Fig 3E). This was also observed in an analysis of *GRP78* and *GLUT1* expression in a micro array of colon normal tissue versus tumor tissue (GSE33114) (Fig S3A) (de Sousa et al, 2011). On the level of protein expression, *A-Grp78$^{WT}$* and *A-Grp78$^{HET}$* organoids also exhibited low levels of GLUT1, potentially contributing to their low levels of glycolysis under normal conditions (Fig 3F). In *AK-Grp78$^{WT}$* organoids however, we found that both GRP78 protein and GLUT1 protein were abundantly expressed (Fig 3F). This supports a potential role for GRP78 as a tumor marker during adenoma progression (Ma et al, 2015; Shen et al, 2019). Heterozygosity for *Grp78* resulted in marked decrease of both GRP78 as GLUT1 protein levels, although expression levels of GLUT1 were still higher than seen in *A-Grp78$^{WT}$* organoids (Fig 3F). These data suggest some, but not full normalization of GLUT1 expression in *AK-Grp78$^{HET}$* organoids compared with *A-Grp78$^{WT}$* organoids.

We next analyzed the expression of GLUT1 protein in a set of 10 paired random samples of human colonic carcinomas with un-known mutation status, with their adjacent normal colonic mucosa (Fig S3B). In seven out of 10 pairs, we observed strongly increased GLUT1 levels in carcinomas compared with their adjacent normal mucosa. These results suggest that GLUT1-mediated glycolysis may be equally important in both large adenomas and carcinomas.

Because reductions in *Grp78* levels may cause nascent proteins to remain inside the endoplasmic reticulum instead of trans-locating to their functional site, we next assessed whether GLUT1 protein could properly localize towards the plasma membrane in *AK-Grp78$^{HET}$* cells (Toyoda et al, 2018). Using flow cytometry on live cells with an antibody directed against GLUT1, we could specifically detect surface GLUT1 (Figs 3G and S4A–C). Consistent with total GLUT1 levels, membrane-localized GLUT1 protein levels were low in *Apc*-mutant organoids regardless of *Grp78* levels, but were

increased on the surface of *AK-Grp78$^{WT}$* cells. Membrane localisation of GLUT1 in *AK-Grp78$^{HET}$* organoids was fully restored to a degree similar to organoids that lacked mutant *Kras*. Thus, although we found total GLUT1 protein to be up-regulated in *AK-Grp78$^{WT}$* and *AK-Grp78$^{HET}$* organoids compared with *A-Grp78* counterparts, *Grp78 heterozygosity* resulted in strongly reduced membrane translocalization of this protein.

To further investigate the functional role of GLUT1 in cellular proliferation, we treated organoids with GLUT1 inhibitor (BAY-876), thereby disrupting glucose uptake and glycolysis (Siebeneicher et al, 2016). We assessed glucose consumption, pyruvate con-sumption, and lactate production as readouts of oxidative phos-phorylation and glycolysis (Figs 3H and I and S5A). Consistent with low levels of GLUT1 expression, *Apc*-mutant organoids exhibited little impact of GLUT1 inhibition in terms of glucose consumption or lactate production (Fig 3H and I). In *AK-Grp78$^{WT}$* organoids, however, GLUT1 inhibition resulted in a considerable dose-dependent re-duction of glucose consumption, lactate production, and prolif-eration (Fig 3H–J). In *AK-Grp78$^{HET}$* organoids, glucose consumption, lactate production, and proliferation were already on a much lower level than *AK-Grp78$^{WT}$*, potentially owing to low levels of surface GLUT1, and additional inhibition of GLUT1 could only marginally decrease growth in *AK-Grp78$^{HET}$* organoids.

As we observed a strong reduction of glycolysis in *AK-Grp78$^{WT}$* organoids upon inhibition of GLUT1, we examined whether GLUT1 inhibition could revert the metabolic switch from oxidative phos-phorylation to glycolysis that is observed in *AK-Grp78$^{WT}$* organoids. Inhibition of GLUT1 decreased the medium lactate to pyruvate ratio in *AK-Grp78$^{WT}$* organoids, suggesting an increase in oxidative phosphorylation when GLUT1 is inhibited (Fig S5B). To understand how GLUT1 inhibition could revert the Warburg effect in *Apc-Kras* mutant organoids, we analyzed whether increased mitochondrial biosynthesis could be responsible for accommodating increased oxidative phosphorylation. Indeed, we found that the level of *Pgc1a*, a transcriptional coactivator that induces mitochondrial biogen-esis, was increased in *AK-Grp78$^{WT}$* organoids upon inhibition of GLUT1, but not in the other organoids (Fig 3K) (Wu et al, 1999).

Altogether, these data indicate that *AK-Grp78$^{WT}$* organoids are skewed towards glycolysis as seen in the Warburg effect and this is impaired when faced with reduced capacity to internalize glucose. GLUT1 inhibition results in reduced glycolysis in *AK-Grp78$^{WT}$* organoids, but a transcriptional activation towards oxidative phosphorylation is suggested. In contrast, *AK-Grp78$^{HET}$* organoids exhibit low levels of glycolysis and blocking glucose uptake via GLUT1 inhibition had little effect on their metabolism. Increased oxidative phosphorylation in these organoids was not observed.

In conclusion, *Grp78* heterozygosity reduces protein and surface expressions of GLUT1 in *Apc-Kras$^{G12D}$* organoids. *Grp78* heterozygosity reduced glycolysis and thus the growth potential of *Apc-Kras$^{G12D}$* organoids by normalizing the enhanced GLUT1 expression in *Kras$^{G12D}$* mutant cells.

## Discussion

The most common initiating mutations of the adenoma to carci-noma sequence, *APC* and *KRAS*, increase proliferation, protein

translation, and stemness. Moreover, these mutations impose metabolic changes that may be critical for tumor growth. *Grp78* is an important regulator of protein-folding capacity and a glucose-sensing protein. Previously, we showed that *Grp78* knockout crypts are not viable, resulting in rapid repopulation by WT crypts, which renders *Grp78* knockout an unviable model for adenoma studies. However, heterozygous targeting of *Grp78* impacts protein translation, stemness, and tumor growth in mice and organoids with an *Apc* mutation (Heijmans et al, 2013; van Lidth de Jeude et al, 2017; van Lidth de Jeude et al, 2018). Here, we explore *Grp78* heterozygosity in mice and organoids with mutations in both *Apc* and *Kras* and we specifically focus on the influence of *Grp78* levels on the additional phenotypical, molecular, and metabolic changes that are caused by a *Kras* mutation in the context of the *Apc* mutant cell state.

We find that adenoma numbers in colons of *AK-Grp78^HET* mice are reduced compared with *AK-Grp78^WT* controls. Although in earlier studies, adenoma reduction was also observed in *A-Grp78^HET* mice compared with *A-Grp78^WT* controls, *AK* animals exhibited adenoma growth primarily in the large intestine and effects of *Grp78* in these animals thus seem specific for animals harboring an additional *Kras* mutation (van Lidth de Jeude et al, 2018).

At the root of reduced colonic adenomas in *AK-Grp78^HET* animals, we find decreased proliferation. Effects that we observe emanate from allelic heterozygosity, but because GRP78-inhibiting compounds have been generated, more forceful therapeutic reduction of GRP78 activity and thereby stronger anti-tumor effects may be feasible without overt toxicities. In preclinical in vitro studies, these compounds have shown reduced proliferation, similar to our findings in *Grp78* heterozygous cells and animals (Hensel et al, 2013; Elfiky et al, 2020).

Ex vivo, in organoids with mutant *Apc* and *Kras*, we find that *Grp78* heterozygosity reduces growth, proliferation, and self-renewal capacity to a level that is similar to organoids with WT *Kras*. We find that in *Apc-Kras* mutant cells, increased proliferation and growth result from increased glucose uptake, compared with *Apc* single-mutant cells, and that these cells switch from oxidative phosphorylation to glycolysis for their energy requirement as is seen during the Warburg effect. This metabolic reprogramming, which is critical for *Apc-Kras* mutant cells is considered a hallmark of cancer and high-*Grp78* levels are required for this switch, because *Grp78* heterozygosity reverts this phenotype (Hanahan & Weinberg, 2011). Glucose uptake in *Apc-Kras* mutant cells is largely mediated by up-regulation of glucose transporter GLUT1 and is reported in several studies to be a prognostic biomarker for worse clinical outcome in multiple malignancies (Yamamoto et al, 1990; Haber et al, 1998). We corroborate earlier findings that *Grp78* plays a central role in the production and localization of GLUT1 (Li et al, 2015; Toyoda et al, 2018). Moreover, we show that *Grp78* is a rate-limiting factor for the Warburg effect as we find that a mild reduction of GRP78 protein level in *Grp78* heterozygous cells already suffices in reducing GLUT1 levels and glycolysis, as opposed to strongly reduced levels as are seen upon knockout or *RNAi*.

Interestingly, mitochondrial respiration was not altered in *AK-Grp78^HET* organoids compared with their WT counterparts which could not be attributed to glucose or pyruvate consumption, suggesting an altered metabolism which would enable these cells

to maintain oxidative phosphorylation at a similar level. Recently, glutamine metabolism was found to play an important role in *Kras*-mutant tumors and perhaps utilization of glutamine could explain the persistence of oxidative phosphorylation in *AK-Grp78^HET* organoids (Wang et al, 2020; Najumudeen et al, 2021). More studies are needed to evaluate the possible synergistic effects of targeting both metabolic pathways as a therapeutic strategy in (colon) cancer.

Taken together, we identify *Grp78* as a critical vulnerability in mutational progression of colorectal tumorigenesis. Our data strengthen the potential of *Grp78* as an attractive target for therapy in CRC.

# Materials and Methods

### Animal experiments

All mouse experiments were performed in the *Academic Medical Center Animal Research Institute* in accordance with local guidelines. *VillinCre^ERT2*, *Apc^fl*, *Kras^G12D*, and *Grp78^fl* alleles were all described previously (el Marjou et al, 2004; Jackson et al, 2001; Luo et al, 2006; Madisen et al, 2010; Sansom et al, 2004; Shibata et al, 1997; Soriano, 1999). All mice had a C57BL/6 background.

For Cre^ERT2-mediated recombination, mice were given daily injections with 1 mg of tamoxifen (T5648; 10 mg/ml in corn oil; Sigma-Aldrich), during five consecutive days. 2 h before euthanizing, all mice received 100 mg/kg BrdU intraperitoneally (10 mg/ml in NaCl; Sigma-Aldrich). The mice were euthanized 8 wk after Cre^ERT2 recombination or earlier (week 6, week 7) when the endpoints, according to the Netherlands Association for Laboratory Animal Science, were reached. Rectal prolapse was a common symptom and accorded as a humane endpoint.

### Tissue preparation and immunohistochemistry

Paraffin embedding and subsequent (immuno)histochemistry was performed as previously described (Heijmans et al, 2011). Primary antibodies that were used are as follows: mutant specific anti-G12D-RAS 1:100 (14429; Cell signaling) (Muthalagu et al, 2020), anti-Grp78 antibody 1:700 (3177S; Cell signaling), anti-BrdU mouse monoclonal 1:500 (BMC9318; Roche), anti-β-catenin 1:2,000 (AB32572; Abcam), and anti-cleaved-caspase-3 1:400 (9661S; Cell Signaling). Antibody binding was visualized with Powervision (Immunologic) and substrate development was performed using diaminobenzidine (D5637-10G; Sigma-Aldrich). Hematoxylin was used as counterstain.

### In situ hybridization

RNAscope was performed according to the manufacturer's protocols. These included the "Formalin-Fixed Paraffin-Embedded Sample Preparation and Pre-treatment for RNAscope 2.5 assay." For in situ assessment of the *Grp78^Δ5−7* mRNA, The BaseScope Reagent Kit (Advanced Cell Diagnostics) was used as previously described (van Lidth de Jeude et al, 2018).

## Organoid culture

Experiments were performed with recombined organoids of four different genotypes, referred to as *A-Grp78$^{WT}$* and *A-Grp78$^{HET}$* and *AK-Grp78$^{WT}$* or *AK-Grp78$^{HET}$*. In organoid experiments, all *Apc* mutations were homozygous as opposed to heterozygosity for *Apc* in mouse experiments.

Crypt harvest and expansion of small intestinal organoid culture were performed as described previously (Heijmans et al, 2013). Recombination of organoids was established by adding 4-hydroxy tamoxifen (4-OHT) (H6278-10MG; Sigma-Aldrich) to culture the medium for 24 h directly after passaging (1 *μM*).

## FACS-based EdU incorporation

We used the Click-iT EdU Alexa Fluor 647 (C10634; Thermo Fisher Scientific) according to the manufacturer's protocol. 72 h after recombination, EdU was added for 4 h. Results were analyzed with FlowJo V10 software.

### *Flow cytometric immunostaining of GLUT1*

72 h after recombination, organoids were harvested. Matrigel was fractionated using a Pasteur pipette and organoids and matrigel were separated by centrifuging. Crypts were detached to single cells by incubating with TrypLE (Gibco) during 10 min at 37°C. Anti-GLUT1 (#15309; Abcam) was added in a 1:500 solution. After incubating and washing, second antibody was added (A21443; Invitrogen). Propidium iodide (PI) was added before processing to flow cytometry to distinguish live (no PI staining) versus dead cells (PI +ve staining because of a broken cell membrane and attached to DNA). Results were analyzed with FlowJo V10 software.

## GLUT1 inhibition

The highly selective GLUT1 inhibitor BAY-876 was dissolved in DMSO (SML 1774; Sigma-Aldrich). Vehicle (DMSO) was compared with concentrations of 10 and 100 nM of BAY-876. After seeding and recombination, medium was changed and vehicle or BAY-876 was added to the crypts for 2 d. For EdU experiments, the medium was changed and both EdUs as BAY-876 or vehicle were added for 4 h before continuing with the manufacturer's protocol (C10634; Thermo Fisher Scientific).

## Measuring global translation rates

For measurement of global protein translation rates, we used previously described measurements in the Materials and Methods section (van Lidth de Jeude et al, 2018). In short, global protein synthesis rates were quantified measuring $^{35}$S-labeled methionine and cysteine into newly translated proteins. Newly translated proteins were shown relative to control samples, after normalization to total protein, using BCA Protein Assay Kit (Pierce).

## RNA isolation

For gene-expression experiments in organoids, *mRNA* isolation was performed 72 h after recombination using the Bioline ISOLATE II

RNA Mini kit (BIO-52073; Bioline) according to the manufacturer's instructions. For *RNA* extraction from mouse intestine, tissue was homogenized with a Miccra D-1 homogenizer in 1 ml Tri-reagent (T9424; Sigma-Aldrich) and *RNA* extraction was performed according to the manufacturer's protocol.

## cDNA synthesis, and quantitative RT–PCR

Synthesis of cDNA was performed using 1 *μg* of purified *RNA* using Revertaid reverse transcriptase according to the protocol (Fermentas). Quantitative RT–PCR was performed using sensiFAST SYBR No-ROX Kit (Bio-98020; GC-biotech) according to the manufacturer's protocol on a Bio-Rad iCycler. Primer sequences were ordered as found on Primer designing tool (https://www.ncbi.nlm.nih.gov/tools/primer-blast/). All primersets were tested and chosen as previously described. Relative gene expression was calculated using the LinReg method. Primers: *36B4* (forward CCAGCGAGGCCACACTGCTG, reverse ACACTG-GCCACGTTGCGGAC), *β-actin* (forward TTCTTTGCAGCTCCTTCGTT, reverse ATGGAGGGGAATACAGCCC), *Grp78* (forward ACTTGGGGACCACCTATTCCT, reverse ATCGCCAATCAGACGCTCC), *Grp78$^{Δ5−7}$* (forward TGGCAC-TATTGCTGGACTGA, reverse TTCAGCTGTCACTCGGAGAA), *Grp94* (forward TTGTGTCCAATTCAAGGTAATCA, reverse TTGCTGACCCAAGAGGAA) *Lgr5* (forward TGTGTCAAAGCATTTCCAGC, reverse CAGCGTCTT-CACCTCCTACC), *Olfm4* (forward AACATCACCCCAGGCTACAG, reverse TGTCCACAGACCCAGTGAAA).

## Human tissue samples

Tissue samples were taken from CRC patients who underwent resection in the Amsterdam UMC, location AMC. The study protocols were approved by the Medical Ethical Committee of the AMC and all patients provided written informed consent. Healthy colonic samples were taken at least 5 cm from the tumor. Samples were homogenized in cell lysis buffer (Cell Signaling Technology) with protease inhibitors (Roche).

## Immunoblot and quantification

Organoids were recombined and lysed after 72 h in a lysis buffer (Cell Signaling) containing Protease Inhibitor Cocktail (#13538100; Roche). Primary antibody detection was performed overnight; antibodies used for detection were anti-GRP78/BiP (#3183, 1:1,000; Cell Signaling), anti-GLUT1 (#15309, 1:1,000; Abcam), and anti-β-actin (A1978, 1:100,000; Sigma-Aldrich). Secondary antibody detection with HRP-labeled polyclonal antibodies was performed (#P0448; Dako, goat anti-rabbit, #P0447; goat anti-mouse, 1:2,000), and visualization was done using Lumilight Plus (12015196001; Roche). Bands were quantified using ImageJ software (version 1.5 ImageJ; NIH).

## Seahorse

Seahorse experiments were performed according to the manufacturer's instructions. 4 d after recombination, organoids were disrupted and transferred to XFe24 cell culture microplates (Agilent). After 2 d in the XFe24cell culture plate, the organoids were

washed twice. The assay medium (Agilent) was added to the wells. For OCR measurements, the compounds oligomycin A (1 μM), FCCP (0.5 μM), rotenone (1 μM) and antimycin A (1 μM) were used. For ECAR, the compounds glucose (10 mM), oligomycin A (1 μM), and 2DG (50 mM) were used. OCR and ECAR were measured in an XFe24 Seahorse machine (Agilent). The experiment was performed and analyzed by the Seahorse software (Wave). Experiments were normalized to total DNA amount.

### Enzymatic assay for glucose, L-lactate, and pyruvate

Enzymatic assays for glucose, L-lactate, and pyruvate were performed as described, using a CLARIOstar microplate reader (BMG LABTECH) (Chang et al, 2021). 24 h after recombination, the medium was refreshed with vehicle or BAY-876 and incubated for 48 h. Glucose was measured using a colorimetric assay with glucose oxidase. L-Lactate was measured with lactate dehydrogenase (Roche). Pyruvate was measured by a homovanilic acid–based fluorometric assay with pyruvate oxidase (P-4591; Sigma-Aldrich). Results were normalized to cell count.

### Array analysis

Publicly available databases were analyzed using R2 platform. A complete description of the bioinformatics tool R2 may be found at https://hgserver1.amc.nl/cgi-bin/r2/main.cgi.

### Statistics

Statistical analysis was performed using GraphPad Prism version 9.1.0 (GraphPad Software, www.graphpad.com). All values are depicted as the mean ± SEM. In experiments comparing two groups, statistical significance was analyzed using $t$ test. For multiple comparisons, one-way or two-way analysis of variance (ANOVA) was used followed by a Bonferroni post-test. All organoid experiments were done in triplicate, with three wells of organoids per condition. Differences were considered statistically significant at $P < 0.05$.

# Supplementary Information

# Acknowledgements

This work was supported by a grant of the Dutch Cancer Foundation (KWF/Alpe 11053/2017-1) and by a grant of the Netherlands Organisation for Scientific Research (NWO-Veni 91615032).

## Author Contributions

CN Spaan: conceptualization, data curation, formal analysis, validation, investigation, visualization, methodology, and writing—original draft, review, and editing.

RJ de Boer: conceptualization, data curation, formal analysis, validation, investigation, visualization, methodology, and writing—original draft, review, and editing.

WL Smit: conceptualization, data curation, validation, and methodology.

JHM van der Meer: data curation, formal analysis, and investigation.

M van Roest: data curation, formal analysis, and methodology.

JLM Vermeulen: data curation, formal analysis, and methodology.

PJ Koelink: resources, data curation, formal analysis, and methodology.

MAJ Becker: data curation, formal analysis, and methodology.

S Go: resources, data curation, formal analysis, and methodology.

J Silva: resources, data curation, formal analysis, and methodology.

WJ Faller: resources, data curation, and methodology.

GR van den Brink: conceptualization, supervision, funding acquisition, and writing—original draft.

V Muncan: conceptualization, supervision, and project administration.

J Heijmans: conceptualization, formal analysis, supervision, funding acquisition, project administration, and writing—original draft, review, and editing.

## Conflict of Interest Statement

The authors declare that they have no conflict of interest.

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
