## [Reviewer comments · Life Science Alliance]

Life Science Alliance

Grp78 is required for intestinal Kras dependent glycolysis proliferation and adenomagenesis

Claudia Spaan, Ruben de Boer, Wouter Smit, Jonathan van der Meer, Manon van Roest, Jacqueline Vermeulen, Pim Koelink, Marte Becker, Simei Go, Joana Silva, William Faller, Vanesa Muncan, Gijs van den Brink, and Jarom Heijmans

DOI: <https://doi.org/10.26508/lsa.202301912>

Corresponding author(s): Ruben de Boer, Amsterdam University Medical Centers and Jarom Heijmans, Amsterdam UMC, University of Amsterdam, Tytgat Institute for Liver and Intestinal Research, Amsterdam Gastroenterology Endocrinology Metabolism

Review Timeline:

Submission Date:	2023-01-09
Editorial Decision:	2023-02-22
Revision Received:	2023-06-22
Editorial Decision:	2023-08-01
Revision Received:	2023-08-09
Accepted:	2023-08-11

Scientific Editor: Novella Guidi

Transaction Report:

February 22, 2023

Re: Life Science Alliance manuscript #LSA-2023-01912-T

Ruben J. de Boer
Amsterdam UMC, University of Amsterdam, Tytgat Institute for Liver and Intestinal Research, Amsterdam Gastroenterology
Endocrinology Metabolism
Gastroenterology and Hepatology
Meibergdreef 69
Amsterdam, Noord Holland 1105BK
Netherlands

Dear Dr. de Boer,

Thank you for submitting your manuscript entitled "Grp78 is required for intestinal Kras-mutation dependent glycolysis proliferation and adenomagenesis" to Life Science Alliance. The manuscript was assessed by expert reviewers, whose comments are appended to this letter. We invite you to submit a revised manuscript addressing the Reviewer comments.

Thank you for this interesting contribution to Life Science Alliance. We are looking forward to receiving your revised manuscript.

Sincerely,

B. MANUSCRIPT ORGANIZATION AND FORMATTING:

Reviewer #1 (Comments to the Authors (Required)):

In this interesting manuscript Spaan et al report data on the function and role of GRP78 in KRAS driven tumorigenesis. They combined the analysis of mouse models with organoid work and analysis of proliferation, stemness and metabolomics in a KRAS mutant background. The work is interesting, but the manuscript contains multiple typographical errors, figures missing/incorrectly labelled thereby hampering its readability. However their data are interesting and add on to the previously reported finding on GRP 78 by their group and others. My specific comments are listed below.

Major comments:

1. Figure 1 A and B. To be able to actually compare the recombination rates, the IHC and base scope analysis should be performed on sequential sections. Can the authors confirm in the same stainings deletion of APC (e.g. nuclear bCatenin accumulation) and mutant KRAS?
2. Is there any phenotypic/histologic difference between the different genotype? For examples stromal accumulation, immune cell infiltration, proliferation, apoptosis?
3. Fig 1C is not an immunoblot, but survival curve. Line 249, fig 1I is not present. At additional spots in the manuscript referral to figures is incorrect.
4. The conclusion is line 255 is too strong; the authors show reduction in (mainly) large adenomas in the colon, but still a high number is present. So there is a role for GRP78, but there is no proof it is crucial, especially with regard to the initiation. What happens if colorectal tumors are engrafted in a GRP78 +/- background? Do they show decreased growth? Alternatively, what happens to the numbers of adenomas in a chemically induced CRC mouse model, representing more early stages of CRC/initiation? This would shed a light on initiation versus progression. There are some indications for that theory in Fig 2 already.
5. Figure 2G, protein loading is not equal based on b-actin, making it difficult to interpret these data and to confirm the increase in GRP78 between A and AK organoids. Again further labelling of the figure is incorrect.
6. Are the organoids derived from the small or large intestine? Staining, especially for OLFM4, since for LGR5 might be technically challenging, would strengthen their conclusion on stemness and the role of GRP78
7. In figure 3E if GLUT1 is crucial and related to KRAS, can the authors explain the significant upregulation upon APC deletion? Which only seems to occur at the mRNA level? Please adjust the Y-axes to start at 0. Does KD or KO of GRP78 result in further decrease GRP78 levels? Alternatively, does a rescue increase GLUT levels?
8. In supplemental figure 3 only 3 samples are shown, while 10 have been tested. Please include the additional samples. The loading control seems overexposed. Furthermore it is unclear if the samples are adenomas or carcinomas. The text indicates carcinomas, while the figure suggests adenomas.
9. For flow cytometry analysis in fig 3G, please show the gating and associated raw data in supplementary figure including the actual number of cells (and not only relative to A-GRP78)
10. Striking to see that although glycolytic act is strongly increased in KRAS mut background, this is not reflected in glucose consumption (Fig 3H). Good to discuss this further.
11. Is increased/alterd GLUT expression also observed in the mouse models they have used?
12. In the discussion, how would the authors envision selective GRP78 therapy for CRC?

Minor comments:

1. For all bar graphs, show individual datapoint (scatter/bar graph), to clearly show experimental number and spread.

Reviewer #2 (Comments to the Authors (Required)):

Spaan et al. examine the involvement of GRP78 in KRAS-driven colorectal cancer. In brief, they find that heterozygous deletion of Grp78 decreases tumour burden in VillinCreER Apcl/fl KrasG12D/+ mice. Using murine organoids, they show that Grp78 deletion compromises protein synthesis, stemness and organoid growth as well as colonogenic capacity. They perform Seahorse analysis on organoids to show that KRAS activation drives a more glycolytic phenotype, which is partially reversed by Grp78 deletion. Given the previous reports, they interrogate GLUT1 expression in the context of Grp78 status and find total and surface levels downregulated upon Grp78 loss. They next perform proliferation and nutrient consumption/release experiments to

show that pharmacological inhibition of GLUT1 reduces glycolysis in Grp78 WT organoids and is less efficient in Grp78-depleted organoids. I think this study provide important new information on the role of GRP78 in KRAS mutant CRC and will be of interest to the field.

1. This study would benefit from a either a better powered in vivo survival experiment, which currently only shows prolapse-free survival (Figure 1C), or from a more in depth analysis of the samples from the experiment they have performed. The authors included 3-5 animals per experimental group, which prevents a proper statistical evaluation of the effects of Grp78 on disease burden. Can the authors use the present data to design a better-powered study?
2. Can more information be provided on GRP78 expression in tumour tissues versus normal tissues, e.g. by IHC in the GEMM (Fig1) and by WB in the patient cohort (FigS3)? Also, can the same preclinical and clinical samples be used to correlate GRP78 and GLUT1 expression?
3. Figure 2H shows that in organoids, KRAS activation results in increased GRP78 expression. Can this be substantiated with human transcriptomic data from publicly available databases? Does Grp78 expression correlate with other prevalent oncogenic events in CRC?
4. In Figure 2, the authors show decreased proliferation/stemness in AK-Grp78-HET versus AK-Grp78-WT organoids, after in vitro recombination. Can the authors show that tumour cell proliferation was also decreased in vivo by analysing tissue sections for proliferation and stem cell markers. Also, were organoid cultures established from tumours and analysed ex vivo for proliferation/stemness?
5. Figure 3A shows that oxidative phosphorylation was not affected in AK-Grp78-HET versus AK-Grp78-WT. However, since Glut1 expression (Fig 3F) and glucose consumption (Fig3H) were markedly decreased, this begs the question which anaplerotic substrate is sustaining TCA activity. Could these cells rely more on glutaminolysis and would Grp78 deletion synergize with inhibition of glutaminase?
6. The authors show that the lactate/pyruvate ratio has decreased in the medium of AK-WT organoids after treatment with a GLUT1 inhibitor and conclude that that cells shift from glycolysis to oxidative phosphorylation. However, the total amount of lactate present in the spent-media may also be affected by the decreased cell proliferation (hence cell number), rather than decreased glycolysis. To unambiguously make a statement on carbon flux, the authors would need to perform a stable isotope tracing experiment with ¹³C-glucose so they would need to reword or perform a new experiment.

Minor comments

1. The authors need to carefully revise the manuscript for correct referencing of the figures. For example, the data mentioned on expression levels of GRP78 in organoids (line 240) are not shown in Fig1C, as stated, but rather in Figure 2G. Consequently, all referencing of the figure panels for Figure1 beyond panel C and for Figure 2 beyond panel G are incorrect.
2. The authors mention 10 patient samples (tumour and normal) were analysed for expression of GRP78, but data are only shown for 3 (FigS3). The full dataset needs to be included.
3. The supplemental figure legends are missing.

Reviewer #3 (Comments to the Authors (Required)):

This is an interesting manuscript which should be published.

Referee #1:

1. Figure 1 A and B. To be able to actually compare the recombination rates, the IHC and base scope analysis should be performed on sequential sections. Can the authors confirm in the same stainings deletion of APC (e.g. nuclear β Catenin accumulation) and mutant KRAS?

In order to increase comparison between recombination rates, we have repeated IHC and ISH on sequential sections (see Figure 1A). Furthermore, we added KRAS and Beta-Catenin immunohistochemistry in order to confirm recombination of the KRAS and Apc allele in experimental animals (see Figure 1A).

2. Is there any phenotypic/histologic difference between the different genotype? For examples stromal accumulation, immune cell infiltration, proliferation, apoptosis?

Although there is extensive research on the crosstalk between epithelial mutations and subsequent effects in stroma, in the current manuscript, we have primarily focused on effects on epithelium and we have used cell culture models that contain epithelial cells only. To further examine the murine phenotype we have added immunohistochemistry for BrdU and cleaved-caspase-3. In AK-Grp78^{HET} animals (Fig 1C, Supplemental figure 1A), although there were no differences in apoptosis, proliferation is reduced compared to AK-Grp78^{WT} control animals.

3. Fig 1C is not an immunoblot, but survival curve. Line 249, fig 1I is not present. At additional spots in the manuscript referral to figures is incorrect.

We apologize for the inaccuracy and have altered the text.

4. The conclusion is line 255 is too strong; the authors show reduction in (mainly) large adenomas in the colon, but still a high number is present. So there is a role for GRP78, but there is no proof it is crucial, especially with regard to the initiation. What happens if colorectal tumors are engrafted in a GRP78 +/- background? Do they show decreased growth? Alternatively, what happens to the numbers of adenomas in a chemically induced CRC mouse model, representing more early stages of CRC/initiation? This would shed a light on initiation versus progression. There are some indications for that theory in Fig 2 already.

We agree with the reviewer that in the used Grp78 heterozygous model, we didn't show complete rescue of adenoma formation in mice, therefore we mitigate our statement in line 255 by removing the word critical.

It would indeed be interesting to further evaluate adenoma formation in additional cancer models with a Grp78 heterozygosity background. In a (chemically) induced adenomagenesis model, the reduction of tumor initiation and growth could be more pronounced, as exemplified by our organoid reseeding studies.

However, the scope of this article was to specifically look into the effect of Grp78 heterozygosity on adenoma formation by Apc and Kras mutations. We agree that this comment is of value for the discussion, thus we added this point in the discussion.

5. Figure 2G, protein loading is not equal based on β -actin, making it difficult to interpret these data and to confirm the increase in GRP78 between A and AK organoids. Again further labelling of the figure is incorrect.

In order to confirm increased protein levels of GRP78 upon *Kras* mutation and reduction in *AK-Grp78^{HET}* animals, we have added additional immunoblot experiments to figure 3. We have altered and thereby corrected figure legends.

6. Are the organoids derived from the small of large intestine? Staining, especially for OLFM4, since for LGR5 might be technically challenging, would strengthen their conclusion on stemness and the role of GRP78

We clarified the type of organoids by adding ‘small intestine’ in the methods and in the figure legends (Fig 2). Indeed, histochemical staining of organoids for OLFM4 or performing *ish* on organoids is challenging. For accurate comparison of stemness in *Grp78^{WT}* with heterozygous organoids we performed quantitative RT-PCR (Figure 2H) for stem cell marker quantification in addition to functional experiments (single cell clonal capacity assays (figure 2I-J).

7. In figure 3E if GLUT1 is crucial and related to KRAS, can the authors explain the significant upregulation upon APC deletion? Which only seems to occur at the mRNA level? Please adjust the Y-axes to start at 0. Does KD or KO of GRP78 result in further diminish GRP78 levels? Alternatively, does a rescue increase GLUT levels?

Indeed, *Glut1* is upregulated in micro-array analysis of *Apc* mutated organoids compared to *wildtype* organoids. We altered figure 3E to visualize that both *Grp78* and *Glut1* are upregulated in organoids with the described mutations.

In the current manuscript we primarily focused on the interplay between *Kras* and *Grp78* and did not extensively investigate *Glut1* expression upon deletion of *Apc* only. To show that the increase of *Glut1* upon deletion of *Apc* is present, but increase is much more upon deletion of both *Apc* and *Kras*, we have added this to the result section in text and a figure (figure 3E). We have added text to cover increase of *Glut1* in several malignancies to the introduction and discussion to place this in perspective and highlight the special role that (malignant) *Ras* signaling holds over *Glut1* increase. For the figure, we have used a 2log scale as is customary in using array data, as is done in many other manuscripts (e.g. Larrayoz et al, Nat Med 2023; Gao et al; Cell 2023).

As towards the suggestions of the reviewer to include knock out or knockdown models, we agree this would be of great interest. In previous manuscripts (Heijmans et al, 2013; van Lidth de Jeude et al, 2017), we have used *Grp78* knockout organoids, but these are not long term viable due to stem cell loss and repopulation from wild type cells. Since in the current manuscript we aimed at performing adenoma studies, that require long term viability of cells, we could not add these models to the current manuscript. We have now explained this in the manuscript discussion.

8. In supplemental figure 3 only 3 samples are shown, while 10 have been tested. Please include the additional samples. The loading control seems overexposed. Furthermore it is unclear if the samples are adenomas or carcinomas. The text indicates carcinomas, while the figure suggests adenomas.

We have put the data of all the samples in figure (supplemental figure 3), furthermore we corrected the labeling. Additionally, we calculated the relative expression of GLUT1 in the carcinoma tissue compared to the normal adjacent tissue using ImageJ software. Unfortunately, the mutation status of the carcinoma tissue samples that we had available are

unknown and we cannot correlate the change in GLUT1 protein level to the harbored mutations. We have added this to the result section of the manuscript.

9. For flow cytometry analysis in fig 3G, please show the gating and associated raw data in supplementary figure including the actual number of cells (and not only relative to A-GRP78)

We have added a novel figure (supplemental figure 4) to show raw data of flow cytometry. We changed the data in raw data and explained this in the Methods section.

10. Striking to see that although glycolytic act is strongly increased in KRAS mut background, this is not reflected in glucose consumption (Fig 3H). Good to discuss this further.

We have unfortunately noticed another error in the labeling of Figure 3H, and thank the reviewer for noticing. Values of glycolysis should have been noted in nanomol and not millimol. We have changed the figure accordingly.

11. Is increased/altered GLUT expression also observed in the mouse models they have used?

We agree with the reviewer that this analysis would be a valuable addition to the manuscript. In past weeks we have tried quantifying GLUT1 by immunohistochemistry, but quantification is unequivocal by means of IHC. Unfortunately we have no murine tissue for immunoblot available for this essay. We therefore rely on organoids that were derived from these mice for quantification of GLUT1 levels.

12. In the discussion, how would the authors envision selective GRP78 therapy for CRC?

We are curious to whether Grp78 directed therapy will enter clinical studies, since we have been confirming a strong anti-malignant effect of Grp78 knockout of heterozygosity in past years. We have entered the notion of using Grp78 directed therapy to the discussion.

Minor comments:

1. For all bar graphs, show individual datapoint (scatter/bar graph), to clearly show experimental number and spread.

For all murine experiments, we have altered the visualization of the graphs that now include both bar graphs and scatterplots. For in vitro experiments, we feel that scatterplots will make the manuscript more difficult to read.

Reviewer #2 (Comments to the Authors (Required)):

1. This study would benefit from a either a better powered in vivo survival experiment, which currently only shows prolapse-free survival (Figure 1C), or from a more in depth analysis of the samples from the experiment they have performed. The authors included 3-5 animals per experimental group, which prevents a proper statistical evaluation of the effects of Grp78 on disease burden. Can the authors use the present data to design a better-powered study?

First of all, we would like to agree with the reviewer that (in retrospect) the experiments would have benefited from larger groups of animals. We are currently still pursuing a number of experiments with the mouse model that we have used, but although we could include the newer mice to the older experiments, we (prior to submission to life sciences) decided against this for the sake of experimental purity. The reason we kept mouse experiments small was a combination of a power calculation that we did for the AmsterdamUMC animal experimental committee and a difficult to breed multi-allele mouse model. With this approach, we could show a significant difference in adenoma numbers. The current study was not powered to detect a survival benefit. Moreover, local regulations prohibit us from performing proper survival experiments when humane endpoints (such as prolapse) are reached. We did add both cleaved caspase3 staining (for apoptosis) as well as BrdU stainings (for proliferation) for more in depth analysis (see Figure 1, Supp. Fig. 1).

2. Can more information be provided on GRP78 expression in tumour tissues versus normal tissues, e.g. by IHC in the GEMM (Fig1) and by WB in the patient cohort (FigS3)? Also, can the same preclinical and clinical samples be used to correlate GRP78 and GLUT1 expression?

In order to clarify GRP78 expression in tumor tissue, we added *Grp78* expression to the microarray of the organoids, as well as a new figure with *GRP78* and *GLUT1* expression in human colonic tissue and colon tumor tissue (public micro array, Supplemental figure 3A). *GRP78* immunohistochemistry is diffusely distributed in the cytoplasm and cell surface of intestinal cells. Therefore, we didn't manage to have an accurate quantification of this staining. We did perform a new western blot of the organoids and show in the same samples *GRP78* and *GLUT1* expression (Fig 3F).

Unfortunately, a new western blot with *GRP78* on the patient cohort samples couldn't be performed, due to shortness of the original samples.

3. Figure 2H shows that in organoids, KRAS activation results in increased GRP78 expression. Can this be substantiated with human transcriptomic data from publicly available databases? Does Grp78 expression correlate with other prevalent oncogenic events in CRC?

We agree with the reviewer that the relation between oncogenic events and *GRP78* expression is of interest. We examined several published microarrays of human CRC samples. Indeed, we do observe upregulated *GRP78* in adenomas/carcinomas compared to normal tissue in vast majority of the arrays (of which we now have included one array as Supplemental figure 3A). In addition, we examined CRC databases to address the question upon which oncogenic mutation *Grp78* upregulation occurs (*KRAS* vs *P53*, *APC* and *BRAF*) but here results are confusing. CRC samples in these databases are mainly whole tumor samples and contamination with non-epithelial cell types may result in difficulties addressing this question between tumor types.

4. In Figure 2, the authors show decreased proliferation/stemness in AK-Grp78-HET versus AK-Grp78-WT organoids, after in vitro recombination. Can the authors show that tumour cell proliferation was also decreased in vivo by analysing tissue sections for proliferation and stem cell markers. Also, were organoid cultures established from tumours and analysed ex vivo for proliferation/stemness?

In order to assess *in vivo* proliferation we have performed immunohistochemistry for BrdU. Indeed proliferation is reduced in *Grp78^{HET}* animals in the non-adenomatous tissue (novel Figure II). We did not find significant reduction of BrdU incorporation within adenomas, but we feel that this data is not unequivocal or reliable due to lesion heterogeneity and difference in fixation of large lesions and thereby differences in quantification of proliferative cells. For the current manuscript, no *ex vivo* experiments were performed using tumor derived organoids.

5. Figure 3A shows that oxidative phosphorylation was not affected in AK-Grp78-HET versus AK-Grp78-WT. However, since Glut1 expression (Fig 3F) and glucose consumption (Fig3H) were markedly decreased, this begs the question which anaplerotic substrate is sustaining TCA activity. Could these cells rely more on glutaminolysis and would Grp78 deletion synergize with inhibition of glutaminase?

We thank the reviewer for these suggestions and added this to the discussion.

6. The authors show that the lactate/pyruvate ratio has decreased in the medium of AK-WT organoids after treatment with a GLUT1 inhibitor and conclude that that cells shift from glycolysis to oxidative phosphorylation. However, the total amount of lactate present in the spent-media may also be affected by the decreased cell proliferation (hence cell number), rather than decreased glycolysis. To unambiguously make a statement on carbon flux, the authors would need to perform a stable isotope tracing experiment with ¹³C-glucose so they would need to reword or perform a new experiment.

We believe this is an interesting and important point for the discussion. We do correct in the assays for the total amount of cells after harvesting the media, therefore we believe we can state we do measure lactate production and pyruvate consumption per cell. We agree with the reviewer that we show indirect proof of oxidative phosphorylation (in the compound assays) by looking at pyruvate consumption and *Pgc1a*. A tracing experiment would be elegant and direct proof of the carbon flux. We have added this to the discussion and mitigated our statement.

Minor comments

1. The authors need to carefully revise the manuscript for correct referencing of the figures. For example, the data mentioned on expression levels of GRP78 in organoids (line 240) are not shown in Fig1C, as stated, but rather in Figure 2G. Consequently, all referencing of the figure panels for Figure1 beyond panel C and for Figure 2 beyond panel G are incorrect.

We agree with the reviewer and we have updated the figures and corrected the referrals throughout the manuscript. We apologize for the inaccuracy of the figure referrals.

2. The authors mention 10 patient samples (tumour and normal) were analysed for expression of GRP78, but data are only shown for 3 (FigS3). The full dataset needs to be included.

We added the complete figure of the western blot in supplemental figure 3. Moreover, in order to facilitate analysis of GLUT1 alterations between carcinoma and normal tissue per patient, we have added relative quantification of GLUT1 expression, corrected by beta-actin to this figure.

3. The supplemental figure legends are missing.

We sincerely apologize for not uploading this document and have added all figures to the current resubmission.

August 1, 2023

RE: Life Science Alliance Manuscript #LSA-2023-01912-TR

Jarom Heijmans
Amsterdam UMC, University of Amsterdam, Tytgat Institute for Liver and Intestinal Research, Amsterdam Gastroenterology
Endocrinology Metabolism
Gastroenterology and Hepatology
Meibergdreef 69
Amsterdam, Noord Holland 1105BK
Netherlands

Dear Dr. Heijmans,

Thank you for submitting your revised manuscript entitled "Grp78 is required for intestinal Kras-mutation dependent glycolysis proliferation and adenomagenesis". We would be happy to publish your paper in Life Science Alliance pending final revisions necessary to meet our formatting guidelines.

- please upload all figure files as individual ones, including the supplementary figure files; all figure legends should only appear in the main manuscript file
- please add a Running Title and a Summary Blurb/Alternate Abstract to our system
- please add ORCID ID for the corresponding author--you should have received instructions on how to do so
- please add the Twitter handle of your host institute/organization as well as your own or/and one of the authors in our system
- please make sure the author order in your manuscript and our system match
- please consult our manuscript preparation guidelines <https://www.life-science-alliance.org/manuscript-prep> and make sure your manuscript sections are in the correct order
- please add an Author Contributions section to your main manuscript text
- please add your main and supplementary figure legends to the main manuscript text after the references section
- please add callouts for Figure S4A-C to your main manuscript text

A. FINAL FILES:

B. MANUSCRIPT ORGANIZATION AND FORMATTING:

Sincerely,

Reviewer #1 (Comments to the Authors (Required)):

Mycomments have been addressed.

Reviewer #2 (Comments to the Authors (Required)):

The authors have improved the manuscript and addressed my concerns.

August 11, 2023

RE: Life Science Alliance Manuscript #LSA-2023-01912-TRR

Mr. Ruben J. de Boer
Amsterdam University Medical Centers
Gastroenterology and Hepatology
Meibergdreef 69
Amsterdam, Noord Holland 1105BK
Netherlands

Dear Dr. de Boer,

Thank you for submitting your Research Article entitled "Grp78 is required for intestinal Kras dependent glycolysis proliferation and adenomagenesis". It is a pleasure to let you know that your manuscript is now accepted for publication in Life Science Alliance. Congratulations on this interesting work.

DISTRIBUTION OF MATERIALS:

Again, congratulations on a very nice paper. I hope you found the review process to be constructive and are pleased with how the manuscript was handled editorially. We look forward to future exciting submissions from your lab.

Sincerely,
